# 18F-Fluorodeoxyglucose Positron Emission Tomography Is Useful in the Evaluation of Prognosis in Retroperitoneal Sarcoma

**DOI:** 10.3390/cancers13184611

**Published:** 2021-09-14

**Authors:** Toru Wakamatsu, Yoshinori Imura, Hironari Tamiya, Toshinari Yagi, Naohiro Yasuda, Sho Nakai, Takaaki Nakai, Hidetatsu Outani, Kenichiro Hamada, Shigeki Kakunaga, Nobuhito Araki, Takafumi Ueda, Satoshi Takenaka

**Affiliations:** 1Department of Orthopaedic Surgery, Osaka University Graduate School of Medicine, 2-2 Yamadaoka Suita, Osaka 565-0871, Japan; y.imura@mc.pref.osaka.jp (Y.I.); hironari.tamiya@oici.jp (H.T.); n.yasuda@ort.med.osaka-u.ac.jp (N.Y.); nakai-sho@mc.pref.osaka.jp (S.N.); h-otani@ort.med.osaka-u.ac.jp (H.O.); hamada-ke@mc.pref.osaka.jp (K.H.); takenaka-sa@mc.pref.osaka.jp (S.T.); 2Department of Musculoskeletal Oncology Service, Osaka International Cancer Institute, 3-1-69 Otemae, Osaka 541-8567, Japan; yagi-to@mc.pref.osaka.jp (T.Y.); n.araki@ashiya-hosp.com (N.A.); 3Department of Orthopaedic Surgery, National Hospital Organization Osaka National Hospital, 2-1-14 Hoenzaka, Osaka 540-0006, Japan; nakai.takaaki.mz@mail.hosp.go.jp (T.N.); kakunaga.shigeki.ya@mail.hosp.go.jp (S.K.); s.uedat@soreiyu.net (T.U.)

**Keywords:** retroperitoneal sarcoma, FDG-PET, SUVmax, dedifferentiated liposarcoma, leiomyosarcoma

## Abstract

**Simple Summary:**

Retroperitoneal sarcomas are difficult malignancies to treat because complete surgical resection is the only effective treatment option, but it is difficult to secure sufficient surgical margins. It is essential for developing a treatment strategy to assess tumor aggressiveness and predict prognosis for patients. However, the aggressiveness of retroperitoneal sarcomas before treatment cannot be fully evaluated. In patients with resectable soft tissue sarcomas or several carcinomas, SUV evaluated with FDG-PET has been reported to be a valuable prognostic parameter. However, the correlation between SUVmax on FDG-PET and the prognosis of several histological subtypes in retroperitoneal sarcoma, including dedifferentiated liposarcoma, well-differentiated liposarcoma, and leiomyosarcoma, remains uncertain. This study revealed that SUVmax calculated with FDG-PET was useful as a prognostic factor in retroperitoneal sarcoma, especially in dedifferentiated liposarcoma and Grade2 retroperitoneal sarcoma.

**Abstract:**

**Background:** Retroperitoneal sarcomas are rare neoplasms that occur in the retroperitoneum. Complete surgical resection is the only effective treatment option. The prediction of prognosis by histological diagnosis has not yet been established. The purpose of this study was to identify the usefulness of [18-F] fluorodeoxyglucose (FDG) positron emission tomography (PET) imaging for validating the prognosis of retroperitoneal sarcoma (RPS) established by histological diagnosis. **Methods:** We retrospectively reviewed 201 patients with RPS treated at the Osaka International Cancer Institute between 2010 and 2021. We extracted the clinical data, including standardized uptake values (SUVs), evaluated with FDG-PET, and statistically analyzed the data. **Results:** The median age of patients was 64 years (range, 31–85 years). A total of 101 (50.2%) patients were men, and 100 (49.8%) were women. Surgical resection was performed in 155 (77.1%) patients. On histological analysis, 75 (37.3%), 52 (25.9%), and 29 (14.4%) patients were diagnosed with dedifferentiated liposarcoma, well-differentiated liposarcoma, and leiomyosarcoma, respectively. The median survival time for patients with high maximum SUV (SUVmax) (≥4) or low SUVmax (<4) was 275.8 months and 79.5 months, respectively. Furthermore, among the patients with dedifferentiated liposarcoma, the overall survival rate for patients with high SUVmax (≥4) was significantly lower than that of those with low SUVmax (<4). **Conclusions:** The present study demonstrated that SUVmax calculated with FDG-PET was useful as a prognostic factor in RPS, especially in dedifferentiated liposarcoma and Grade2 RPS. To devise a treatment strategy for RPS, SUVmax during FDG-PET scan may be considered for clinical assessment.

## 1. Introduction

Retroperitoneal sarcoma (RPS) is a rare mesenchymal tumor consisting of approximately 10% of all soft tissue sarcomas (STS) [1,2,3]. There are many histological subtypes of RPS; approximately 75–80% of the tumors are diagnosed as dedifferentiated liposarcoma (DDLPS), well-differentiated liposarcoma (WDLPS), or leiomyosarcoma (LMS) [4,5]. Although complete resection by surgery is the only effective treatment to improve prognosis, it is difficult to secure sufficient surgical margins because of the tumor size and location of the RPS [6,7]. In contrast, chemotherapy and radiotherapy do not play a pivotal role in the treatment strategy for RPS [8,9]. Recently, a randomized clinical trial (EORTC-62092) failed to demonstrate the benefit from preoperative radiotherapy for RPS, except for the liposarcoma subgroup [9]. Thus far, unlike soft tissue sarcoma of the extremities, multimodality treatment of surgical resection combined with chemotherapy and/or radiotherapy for RPS has not been established. The most beneficial treatment strategy for RPS might be repeated excision for primary and recurrent or metastatic tumors, and chemotherapy and/or radiotherapy are considered effective when the tumors become unresectable [6].

To develop a treatment strategy and predict prognosis for patients with malignant tumors, assessment of tumor aggressiveness is essential, determined by pathological diagnosis and Grade. Furthermore, the clinical tumor stage and size are evaluated, and then the treatment strategy is designed. However, the aggressiveness of RPS before treatment cannot be fully assessed because a biopsy is often not even performed. However, if a biopsy was performed, the biopsy tissue sometimes does not reflect the tumor aggressiveness because of the tumor size and heterogeneity of RPS. Various histological subtypes of RPS are often treated similarly without appropriate clinical assessment.

Positron emission tomography (PET) using 2-fluoro-2-deoxy-D-glucose (FDG) is an imaging tool clinically used for early detection of primary and metastatic tumors with high sensitivity by evaluating tumor glucose metabolism [10,11]. Standardized uptake values (SUVs) reflecting FDG uptake by tumors can be calculated. In patients with resectable STS, SUV has been reported to be a valuable prognostic parameter similar to several carcinomas, including lung, colon, and pancreas [12,13]. However, the utility of PET for predicting the prognosis of RPS is not clearly understood. A study reported that histologic subtype and Federation Nationale des Centres de Lutte le Cancer (FNCLCC) Grade in patients with retroperitoneal liposarcoma could be assessed by the maximum standardized uptake value (SUVmax) measured on FDG-PET [14]. Although another study showed that SUVmax on FDG-PET improved outcome prediction in retroperitoneal liposarcoma, it did not reveal that SUVmax was a valuable prognostic factor because all histological subtypes of liposarcomas in RPS were included and equally analyzed with the same stats [15]. WDLPS is known to have low malignant potential, but DDLPS is a high-grade malignancy. To date, the correlation between SUVmax on FDG-PET and the prognosis of DDLPS in RPS have not been clearly demonstrated. Likewise, the correlation between SUVmax on FDG-PET and the prognosis of other histological subtypes in RPS remains uncertain.

This study proposed that FDG-PET is a good tool for evaluating the early detection of tumors and tumor aggressiveness in RPS. We aimed to clarify that SUVmax measured with FDG-PET predicted the aggressiveness and prognosis of DDLPS and other histological subtypes in RPS. We also investigated the involvement of SUVmax in the clinical outcome of RPS patients compared with other candidate prognostic factors.

## 2. Materials and Methods

Retroperitoneal tumors were detected in 252 patients between 2010 and 2021 at the Osaka International Cancer Institute. We excluded 51 patients in this retrospective study who were lost to follow-up without treatment or diagnosed with benign tumors, including lipoma or schwannoma. As a result, 201 patients with RPS were recruited and analyzed in the current study. We also included patients who were clinically diagnosed with retroperitoneal malignant tumors without biopsy for pathological diagnosis. We retrospectively reviewed their medical records. The data of 201 patients with RPS, including their clinicopathological characteristics (age, gender, metastasis at the first visit, pathological histology, tumor grade, tumor size, and SUVmax with FDG-PET), treatment modalities (surgical resection, surgical margin, chemotherapy, and radiotherapy), and clinical outcomes were investigated (Table 1). Musculoskeletal pathologists diagnosed almost all cases of RPS at our institute. FDG-PET was performed for 145 patients at our institute or referral hospitals. We extracted the data with the highest SUVmax, and, in almost all cases, there was no significant difference in the SUVmax for each patient during their observation periods.

The median age of the patients was 64 years (range, 31–85 years). The analyzed patients consisted of 101 men and 100 women. The median follow-up period was 41.8 months (range, 0.8–293.1 months). Distant metastasis was observed in 13 patients with RPS at the first visit to the hospital. The histological diagnoses were as follows: 75 patients with dedifferentiated liposarcoma (DDLPS), 52 with well-differentiated liposarcoma (WDLPS), 29 with leiomyosarcoma (LMS), and 45 with other histological types of sarcoma, including undifferentiated pleomorphic sarcoma, myxoid liposarcoma, myxofibrosarcoma, and solitary fibrous tumor. For tumor grade, 67, 90, and 34 patients were diagnosed with Grade1, Grade2, and Grade3 tumors, respectively. The tumor size was less than 5 cm in 37 patients, between 5 cm and 15 cm in 90 patients, and 15 cm or more in 66 patients. Surgical resection was performed in 155 patients with R0–1 or R2 resection in 80 or 40 patients, respectively. Thirty-five patients’ surgical margins were unknown; chemotherapy as neoadjuvant, adjuvant, or advanced stage therapy was administered to 85 patients, and radiotherapy was administered to 56 patients. The median SUVmax was 4 (range, 0–25.5). A summary of these clinicopathological and treatment characteristics is shown in Table 1.

We compared the overall survival rate (OS) or disease-free survival rate (DFS) of each group using the Kaplan–Meier method and log-rank test for statistical estimates in univariate analysis. OS was defined as the time from the initial presentation to death associated with the disease or the last follow-up. DFS was defined as the time from the first surgical resection to the first detection of local recurrence, distant metastasis, or the last follow-up. We excluded patients with distant metastasis at the initial referral or without definitive surgery from the DFS analysis. Multivariate analysis was performed using the Cox proportional hazard method with variables. Statistical significance was set at *p* < 0.05. Statistical analyses were performed using EZR software (Saitama Medical Center, Jichi Medical University, Saitama, Japan), a graphical user interface for R (The R Foundation for Statistical Computing, Vienna, Austria). Ethical approval for this study was obtained from the institutional review board of the Osaka International Cancer Institute.

## 3. Results

The 5-year and 10-year OS rates of all 201 patients with RPS were 74.7% and 59.8%, respectively. According to the histological subtypes, the 5-year OS rates were 94.5%, 70.8%, 67.8%, and 59.7%, and the 10-year OS was 90.6%, 48.6%, 44.5%, and 45.5% for WDLPS, DDLPS, LMS, and other, respectively (Figure 1). The prognosis of patients with DDLPS, LMS, and other subtypes was similarly worse than those with WDLPS. On the other hand, the 5-year DFS rates were 35.9%, 19.9%, 20.5%, and 42.2% for WDLPS, DDLPS, LMS, and others, respectively. These results were considered to reflect a high local recurrence rate in DDLPS and frequent distant metastasis in LMS compared to WDLPS and others.

We used the SUVmax threshold of 4 because this threshold was the median score of all patients with RPS in this study, and the prognosis of each group was the most significant with this setting. The 5-year OS of the SUVmax low (<4) and SUVmax high (≥4) groups were 92.9% and 58.4%, respectively, showing that the SUVmax was a strong prognostic factor for RPS (*p* < 0.001, Figure 2a). Furthermore, the 5-year DFS rates of the SUVmax low and SUVmax high groups were 34.5% and 12.7%, respectively (*p* = 0.027, Figure 2b). In addition, we compared the characteristics of the patients in the low- and high SUVmax groups. Most of the patients with Grade1 tumors had low SUVmax (83.3%, 35 out of 42), and most of the patients with Grade3 tumors had high SUVmax (92.6%, 25 out of 27). However, patients with Grade2 tumors were recruited in both SUVmax low and SUVmax high groups (Table 2). After that, we compared patient prognosis in the SUVmax low and SUVmax high groups by stratification with tumor Grade. It showed that patients with high SUVmax and Grade2 tumors had poor prognosis compared to those with low SUVmax and Grade2 tumors, and patients with high SUVmax and Grade3 tumors showed the poorest prognosis. There were insufficient numbers of patients with low SUVmax and Grade3 tumors or high SUVmax and Grade1 tumors for the abovementioned comparisons. (Figure 2c). These results suggested that SUVmax was a strong prognostic factor of at least Grade2 tumors in RPS.

We also performed analyses of OS with other variables using univariate analysis, which showed that gender, metastasis as the first visit, tumor Grade, histology, tumor size, surgical resection, chemotherapy, and radiotherapy were significant prognostic factors in RPS (Table 3). Similar to previous studies, patients who received chemotherapies or radiotherapies had a worse prognosis than those who did not receive these treatments, suggesting that the benefit of chemotherapy or radiotherapy was small for RPS (Table 3) [8,9]. Multivariate analyses demonstrated that a high SUVmax (hazard ratio [HR] = 9.871; 95% confidence interval [CI]: 2.855–34.13; *p* < 0.001) was independently correlated with shorter OS (Table 4). Surgical resection (hazard ratio [HR] = 0.165; CI: 0.06428–0.4237; *p* < 0.001) was also an independent prognostic factor for RPS (Table 4). Tumor Grade (hazard ratio [HR] = 2.49; 95% CI: 0.505–12.28; *p* = 0.2624) was not an independent prognostic factor. Meanwhile, in multivariate analyses, SUVmax was not independently correlated with DFS.

It was suggested that SUVmax reflected aggressiveness and was useful for the prognostic implication of RPS. However, more patients with WDLPS or Grade1 tumors were recruited in the SUVmax low group than in the SUVmax high group (Table 2). Generally, tumor grade is a strong prognostic factor, and WDLPS is a low-grade malignant tumor. SUVmax in WDLPS (median, 1.9; range, 0–15.6) was lower than that in DDLPS (median, 5.7; range, 0–25.5), LMS (median, 6.75; range, 1.8–20.6), and other subtypes (median, 6.05; range 0–25.5) (Figure 3). Therefore, we investigated the correlation between SUVmax and RPS prognosis by histological subtype.

We classified patients with RPS into four groups according to histological subtypes: WDLPS, DDLPS, LMS, and others. Regarding OS, univariate analysis revealed that high SUVmax was correlated with a shorter prognosis in the DDLPS and other groups (Figure 4a–c and Appendix A, Table 5 and Appendix A). No patient with WDLPS died during the observation period, and the prognosis of SUVmax high tended to be shorter on LMS but was not statistically significant (Figure 4b,c, Table 6). In addition, DFS was shorter only for patients with high SUVmax with LMS. Multivariate analyses indicated that SUVmax and surgical resection were independent prognostic factors in patients with DDLPS, but not those with other subtypes (Table 7 and Appendix A).

We also compared the characteristics of the SUVmax low or SUVmax high groups in DDLPS. Most of the tumors were Grade2 in the SUVmax low group, but 45.5% (15 out of 33) tumors were Grade3 with high SUVmax (Table 8). Since tumor Grade was a strong prognostic factor, we hypothesized that a short prognosis in patients with high SUVmax resulted from Grade3 tumors. On the contrary, the prognosis of patients with high SUVmax and Grade2 tumors were worse than those with low SUVmax and Grade2 tumors (Figure 5). In addition, there was no significant difference in the prognosis of patients with Grade2 and Grade3 tumors in the SUVmax high group (Figure 5). These results indicated that SUVmax had the potential clinical usefulness in better assessing the prognosis for RPS patients with DDLPS and Grade2 tumors.

In conclusion, the current study demonstrated that SUVmax measured with FDG-PET was useful in determining the prognostic value of RPS, especially in DDLPS and other groups, but not in LMS. Since SUVmax with FDG-PET reflected the prognosis of patients with DDLPS and Grade2 RPS prior to FNCLCC Grade, FDG-PET may be considered an essential tool for clinical assessment in RPS.

## 4. Discussion

The present study demonstrated that a high SUVmax threshold (≥4) was a strong prognostic factor for RPS, particularly DDLPS and Grade2 RPS tumors. Recently, Sambri et al. demonstrated that sarcoma patients with SUVmax of less than 10.3 showed a better prognosis and low recurrence rate in some histological types [16]. Subramaniam et al. also presented an association of higher SUVmax with higher histological Grade, worse recurrence-free survival, and poor OS in RPS, including DDLPS and LMS [17].

In the past, nomograms were created for primary and recurrent RPS to evaluate the patients’ OS or DFS [4,5,18,19]. The nomogram aided in the prediction of OS or DFS probability corresponding to a combination of patient’s covariates as follows: tumor size, FNCLCC Grade, histological subtype, multifocality, age, presentation (primary or recurrent), complete resection of surgery, chemotherapy or radiotherapy after the first surgery, and the number of resected organs at the first surgery. However, SUVmax with FDG-PET was not included in the nomogram. Our investigation demonstrated that SUVmax was correlated with the prognosis in Grade2 RPS and DDLPS. Brenner et al. reported SUVmax obtained by FDG-PET was a more useful parameter for risk assessment in liposarcoma than tumor Grade [20]. Therefore, SUVmax for Grade2 RPS and DDLPS may be considered an additional parameter within the nomogram, resulting in prognostic implications that could help decision-making regarding treatment intensity, such as combined resection of organs for complete resection of surgery, dose intensity, or regimens for chemotherapy or radiotherapy.

On the other hand, LMS was one of the most frequent malignant tumors in the retroperitoneum, and we could not suggest a clear prognostic difference between SUVmax low and SUVmax high in LMS patients. Since LMS showed high metastatic potential but not recurrence in our experience, many patients were lost to follow-up without treatment. There was a huge difference in the treatment strategy for each doctor in our institution, resulting in biased data on LMS.

In clinical practice, radical cure of RPS is difficult because the tumors are often too large to perform complete resection with wide margins, such as soft tissue sarcoma in the extremities. A previous study demonstrated histopathologic organ invasion in approximately 25% of adherent organs, even when not suspected intraoperatively [21]. Although combined resection of organs was needed to complete resection in many cases of RPS, it was reported that there was a higher operative risk in surgery involving pancreaticoduodenectomy, major vascular resection, and splenectomy/pancreatectomy, which might cause a potential treatment conflict [22]. Consequently, novel treatments or drugs that can shrink retroperitoneal tumors and prevent organ invasion are required for the radical cure of RPS.

Our data showed that RPS, at least DDLPS, with high SUVmax had a poor prognosis, indicating that cancer metabolism strongly links with tumor aggressiveness in RPS. Thus, targeting the metabolism of RPS might be promising for novel treatment strategies, and many studies have reported the anti-tumor effect of metabolic targeting in several sarcomas, including liposarcoma and Ewing sarcoma [23,24,25,26]. In most cases of DDLPS and WDLPS, gene amplification of MDM2 proto-oncogene (MDM2) and cyclin-dependent kinase 4 (CDK4) has been detected [27,28,29,30]. Based on our data, this might be a new strategy for seeking effective treatment targets from additional gene alterations in high SUVmax tumors than those with low SUVmax.

The present study has several limitations. First, this was a retrospective study using the clinical data of a single institute. Therefore, it is possible that unintentional bias in the selection of patients could not be fully eliminated. Second, the number of patients was too small to perform sufficient statistical analyses for several histological subtypes of RPS, such as LMS or undifferentiated pleomorphic sarcoma.

In conclusion, we demonstrated that FDG-PET imaging is useful in evaluating prognosis, and SUVmax reflects tumor aggressiveness in RPS, particularly DDLPS. We propose that FDG-PET may be considered before developing a treatment strategy for RPS.

## 5. Conclusions

The prognosis of retroperitoneal sarcoma patients with high maximum SUV (SUVmax) (≥4) was worse compared to the patients with low SUVmax (<4). Furthermore, among the patients with dedifferentiated liposarcoma or Grade2 retroperitoneal sarcoma, the overall survival rate for patients with high SUVmax (≥4) was significantly lower than those with low SUVmax (<4). These data demonstrated that SUVmax calculated with FDG-PET was useful as a prognostic factor in retroperitoneal sarcoma, especially in dedifferentiated liposarcoma and Grade2 retroperitoneal sarcoma. SUVmax on FDG-PET scan may be considered for clinical assessment to develop a treatment strategy for retroperitoneal sarcoma.

## Figures and Tables

**Figure 1 cancers-13-04611-f001:**
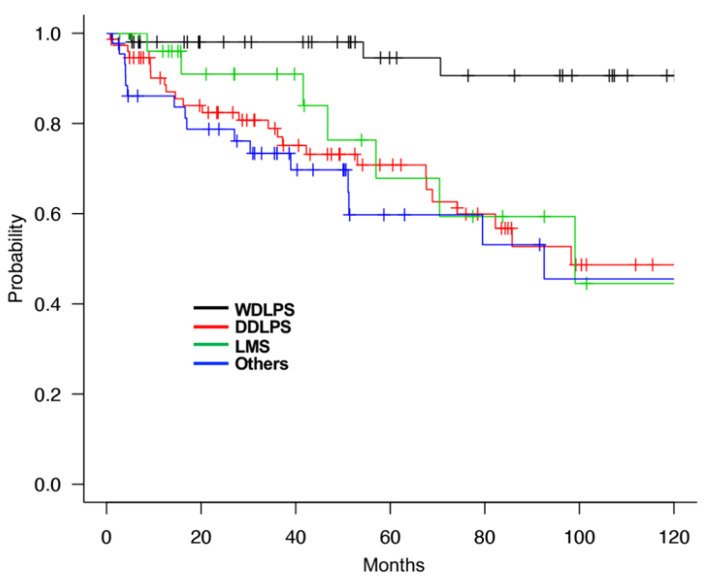
Overall survival (OS) of RPS patients by histological subtypes. WDLPS; well-differentiated liposarcoma, DDLPS; dedifferentiated liposarcoma, LMS; leiomyosarcoma.

**Figure 2 cancers-13-04611-f002:**
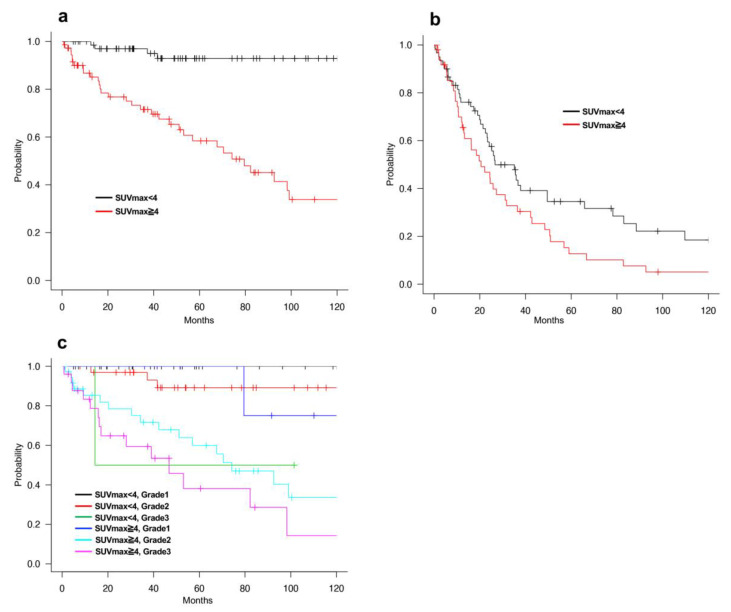
(**a**) OS of all RPS patients with SUVmax low (<4) or SUVmax high (≥4). (**b**) DFS of all RPS patients with SUVmax low (<4) or SUVmax high (≥4). (**c**) OS of RPS patients with SUVmax low and SUVmax high group by stratification with tumor grade.

**Figure 3 cancers-13-04611-f003:**
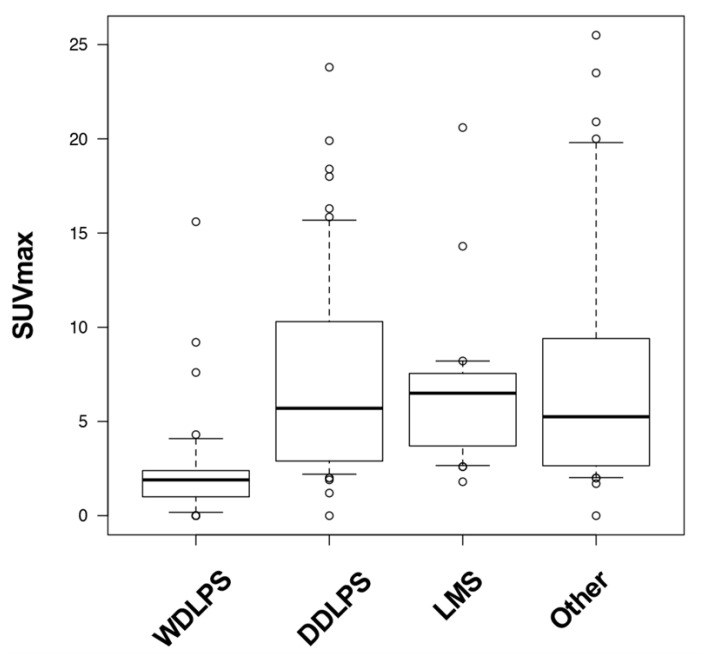
Distribution of SUVmax by histologic subtype.

**Figure 4 cancers-13-04611-f004:**
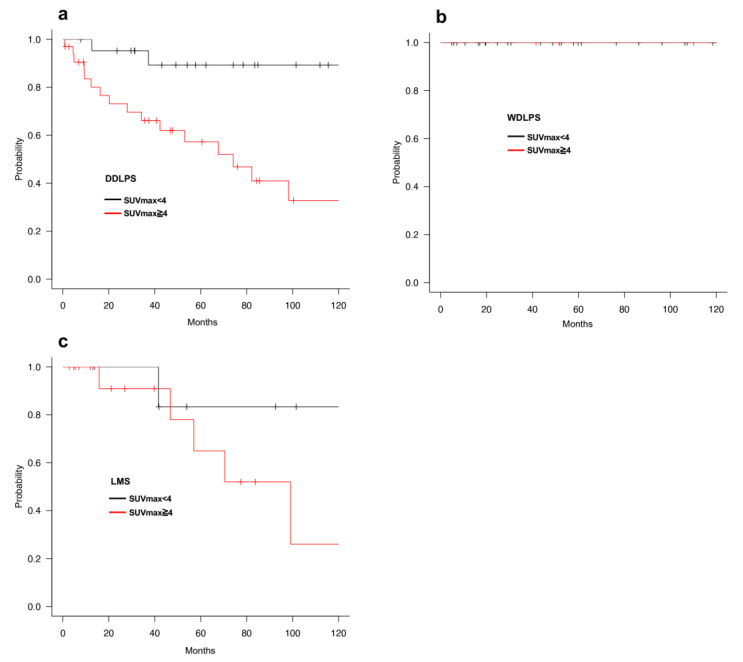
(**a**–**c**) OS of RPS patients with SUVmax low (<4) or SUVmax high (≥4) by histological subtypes, (**a**); DDLPS, (**b**); WDLPS and (**c**); LMS.

**Figure 5 cancers-13-04611-f005:**
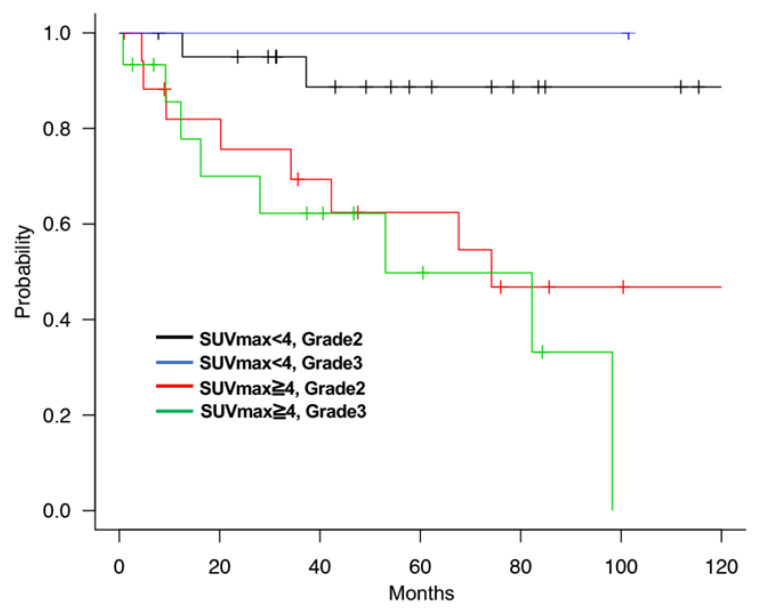
OS of DDLPS patients with SUVmax low and Grade2, SUVmax high and Grade2 and SUVmax and Grade3.

**Table 1 cancers-13-04611-t001:** Clinicopathologic and treatment characteristics.

Characteristic		No. (%)
Age, years	Range	31–85
	Median	64
Gender	Male	101 (50.2)
	Female	100 (49.8)
Metastasis at the first visit	Yes	13 (6.5)
	No	188 (93.5)
Histology	Dedifferentiated liposarcoma	75 (37.3)
	Well-differentiated liposarcoma	52 (25.9)
	Leiomyosarcoma	29 (14.4)
	Other	45 (22.4)
Tumor grade	Grade1	67 (33.3)
	Grade2	90 (44.8)
	Grade3	34 (16.9)
	NE	10 (5.0)
Tumor size	<5 cm	37 (18.4)
	≥5 cm and <15 cm	90 (44.8)
	≥15 cm	66 (32.8)
	NE	8 (4.0)
Surgical resection	Yes	155 (77.1)
	R0 and R1	80 (51.6)
	R2	40 (25.8)
	Unknown	35 (22.6)
	No	46 (22.9)
Chemotherapy	Yes	85 (42.3)
	No	116 (57.7)
Radiotherapy	Yes	56 (27.9)
	No	145 (72.1)
SUVmax	Range	0–25.5
	Median	4
NE; not estimated		

**Table 2 cancers-13-04611-t002:** Clinicopathologic and treatment characteristics between SUVmax low and SUVmax high.

Characteristic	SUVmax Low (*n* = 72)	SUVmax High (*n* = 73)	*p* Value
Age, years (Range, median)	34–83, 63	31–85, 63	0.814
Gender (male/female)	26/46	45/28	0.002
Localized/metastatic	70/2	63/10	0.017
Histology			<0.001
Well-differentiated liposarcoma	29	4	
Dedifferentiated liposarcoma	22	32	
Leiomyosarcoma	7	17	
Other	14	19	
Tumor grade			<0.001
Grade1	35	7	
Grade2	33	37	
Grade3	2	25	
NE	2	4	
Tumor size			0.961
<5 cm	17	13	
≥5 cm	54	59	
NE	1	1	
Surgical resection (Yes/No)	60/12	51/22	0.132
Chemotherapy (Yes/No)	25/47	45/28	0.007
Radiotherapy (Yes/No)	59/13	31/42	0.03
NE; not estimated			

**Table 3 cancers-13-04611-t003:** Analyses on survival in all RPS patients.

Variable			Univariate Analysis		
		No.	5 Year-OS (%)	95% CI (%)	*p* Value
Age	60>	78	76.1	62.2–85.4	0.196
	60≤	123	73.7	63.1–81.7	
Gender	Male	101	67.5	55.2–77.2	<0.001
	Female	100	81.4	70.1–88.8	
Metastasis at first visit	Yes	13	41.5	12–69.5	<0.001
	No	188	76.8	68.5–83.1	
Tumor Grade	1	67	95.3	81–98.9	<0.001
	2 and 3	124	62.4	51.5–71.5	
Histology	WDLPS	52	94.5	78.8–98.7	<0.001
	DDLPS, LMS and Others	149	67.4	57.4–75.6	
Tumor size	<5 cm	37	87.6	65.9–95.9	0.0342
	≥5 cm	156	72.4	63.2–79.8	
Surgical resection	Yes	155	80.5	71.8–86.7	<0.001
	No	46	52.4	32.2–69.1	
Chemotherapy	Yes	85	63.7	51.1–73.9	<0.001
	No	116	85.6	75.6–91.7	
Radiotherapy	Yes	56	64.4	49–76.3	<0.001
	No	145	79.9	70.4–86.7	
SUVmax	Low (4>)	72	92.9	82–97.3	<0.001
	High (4≤)	73	58.4	44–70.3	

**Table 4 cancers-13-04611-t004:** Risk factor for OS in all RPS patients.

Covariates	HR (95% CI)	*p* Value
Age (60> vs. 60≤)	0.9331 (0.4439–1.962)	0.8551
Gender (Male vs. Female)	0.6554 (0.3642–1.789)	0.5982
Metastasis at first visit (Yes vs. No)	2.568 (0.8083–8.162)	0.1098
Tumor Grade (1 vs. 2 and 3)	2.49 (0.505–12.28)	0.2624
Histology (WDLPS vs. DDLPS, LMS, and Other)	10.33 (0.5766–184.9)	0.1127
Tumor size (<5 cm vs. ≥5 cm)	1.948 (0.7235–5.243)	0.187
Surgical resection (Yes or No)	0.165 (0.06428–0.4237)	<0.001
Chemotherapy (Yes or No)	0.8073 (0.3642–1.789)	0.5982
Radiotherapy (Yes or No)	1.009 (0.459–2.218)	0.9824
SUVmax (4> vs. 4≤)	9.871 (2.855–34.13)	<0.001

**Table 5 cancers-13-04611-t005:** Analyses on survival in all DDLPS patients.

Variable			Univariate Analysis		
		No.	5 Year-OS (%)	95% CI (%)	*p* Value
Age	60>	22	79.9	54.8–92	0.176
	60≤	53	67.1	50.2–79.3	
Gender	Male	46	69.1	51.7–81.3	0.131
	Female	29	75.5	53–88.3	
Metastasis at first visit	Yes	2	NA	NA	<0.001
	No	73	71.3	58.1–81.6	
Tumor Grade	2	53	72.9	56.8–83.7	0.219
	3	19	60.8	31.7–80.6	
Tumor size	<5 cm	11	83.3	27.3–97.5	0.123
	≥5 cm	63	67.9	53.2–78.9	
Surgical resection	Yes	65	77.6	63.6–86.8	<0.001
	No	10	NA	NA	
Chemotherapy	Yes	37	58.9	39.5–74.0	0.17
	No	38	85.4	68.3–74	
Radiotherapy	Yes	28	59.2	37.1–75.8	0.00346
	No	47	79.9	63.5–89.5	
SUVmax	Low (4>)	22	89.3	63.2–97.2	<0.001
	High (4≤)	33	57.2	36.6–73.4	

**Table 6 cancers-13-04611-t006:** Analyses on survival in all LMS patients.

Variable			Univariate Analysis		
		No.	5 Year-OS (%)	95% CI (%)	*p* Value
Age	60>	18	49.9	16.5–76.5	0.12
	60≤	11	100	NA	
Gender	Male	8	100	NA	0.795
	Female	21	61.2	28.8–82.5	
Metastasis at first visit	Yes	3	100	NA	0.441
	No	26	65.2	33.6–84.6	
Tumor Grade	2	19	72.3	34.3–90.7	0.667
	3	8	41.7	1.1–84.3	
Tumor size	<5 cm	5	66.7	5.4–94.5	0.551
	≥5 cm	21	72.6	32–91.4	
Surgical resection	Yes	24	67.6	37.2–99.1	0.768
	No	5	NA	NA	
Chemotherapy	Yes	15	41.8	10.2–71.6	0.0231
	No	14	100	NA	
Radiotherapy	Yes	7	50	5.8–84.5	0.241
	No	22	73.1	34.7–91.1	
SUVmax	Low (4>)	7	83.3	27.3–97.5	0.244
	High (4≤)	17	64.9	24.9–87.4	

**Table 7 cancers-13-04611-t007:** Multivariate Cox regression model of OS in DDLPS.

Covariates	Multivariate Analysis	
	HR (95% CI)	*p* Value
Age (60> vs. 60≤)	2.022 (0.6103–6.697)	0.2494
Gender (Male vs. Female)	0.6674 (0.2305–1.932)	0.4558
Metastasis at first visit (Yes vs. No)	20.41 (0.7559–551.2)	0.07288
Tumor Grade (2 vs. 3)	1.085 (0.3348–3.516)	0.8918
Tumor size (<5 cm vs. ≥5 cm)	8.129 (0.7731–85.47)	0.08088
Surgical resection (Yes or No)	0.09174 (0.02071–0.4064)	0.001656
Chemotherapy (Yes or No)	1.058 (0.3098–3.614)	0.9282
Radiotherapy (Yes or No)	1.686 (0.5881–4.832)	0.3311
SUVmax (4> vs. 4≤)	6.056 (1.257–29.19)	0.0248

**Table 8 cancers-13-04611-t008:** Clinicopathologic and treatment characteristics between SUVmax low and SUVmax high in DDLPS.

Dedifferentiated Liposarcoma			
Characteristic	SUVmax Low (*n* = 22)	SUVmax High (*n* = 33)	*p* Value
Age, years (60>/60≤)	7/15	11/22	0.909
Gender (male/female)	11/11	24/9	0.089
Localized/metastatic	22/0	31/2	0.247
Tumor grade			0.001
Grade1	0	0	
Grade2	21	18	
Grade3	1	15	
NE	0	0	
Tumor size			0.756
<5 cm	4	6	
≥5 cm	18	27	
NE	0	0	
Surgical resection (Yes/No)	21/1	27/6	0.142
Chemotherapy (Yes/No)	7/15	22/11	0.011
Radiotherapy (Yes/No)	4/18	15/18	0.038

## Data Availability

The data presented in this study are available on request from the corresponding author. The data are not publicly available because the Institutional Ethics Committee did not provide a specific authorization.

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
