# Peer review of "18F-Fluorodeoxyglucose Positron Emission Tomography Is Useful in the Evaluation of Prognosis in Retroperitoneal Sarcoma"

_cancers, 2021, doi:10.3390/cancers13184611_

Round 1

Reviewer 1 Report

This MS presents an overall very nice analysis with an exciting finding.

There are a few issues to address: 1.) the author state (line 102) that margins can not be reported because of lack of surgical records. This is a no go, and without this information, this paper should not be published. However, apart from the fact that it is strange that surgical records are missing, this information has to be found in the path records. Because the pathologically determined margin is anyway more important than the surgeon's, this query can be addressed in the revision.

2.) on line 230 and following, the authors state that authors state that SUVmax correlated stronger that tumor grade in the prognosis of DDLS. This is a strong statement, however, this reviewer may have missed the information proving this. Fig 5 does not imply seem to prove this.

3.) the authors mention also the seemingly superiority of the SUVmax as opposed to the widely accepted nomogram. It is kindly suggested to put the own results into more gently perspectives. SUVmax for DDLS may be considered as additional parameter within the nomogram etc. But not more. After all, the authors have NOT shown the results for the stated superiority.

4.) the expression "should be considered" in the abstract as well as in the conclusion is too strong. It is suggested to use "may be considered".

Author Response

Dear Reviewer1

Thank you very much for reviewing our manuscript and offering valuable advice.

We have addressed your comments with point-by-point responses and revised the manuscript accordingly.

I attached the table of responses for the comments.

Comment

Response

Revised point

1.) the author state (line 102) that margins cannot be reported because of lack of surgical records. This is a no go, and without this information, this paper should not be published. However, apart from the fact that it is strange that surgical records are missing, this information has to be found in the path records. Because the pathologically determined margin is anyway more important than the surgeon's, this query can be addressed in the revision.

Thank you for your kind advise. Surgical margin is very important for oncological treatment. We re-checked and evaluated surgical and pathological records. However, some surgical margins were not determined, because their margins were not clearly indicated, or the operations were performed in other hospitals. We described unclear margin as “Unknown".

We added the data of surgical margin in Table 1 and the sentence in line 121 and 139-140 and deleted the sentences in line 141.

2.) on line 230 and following, the authors state that SUVmax correlated stronger that tumor grade in the prognosis of DDLS. This is a strong statement, however, this reviewer may have missed the information proving this. Fig 5 does not imply seem to prove this.

Thank you for your indication, and I am very sorry to have caused your misunderstanding. In Figure 5, we demonstrated that grade3 DDLS with SUVmax<4 showed better prognosis than that of grade2 DDLS with SUVmax>4. And mutivariate analysis in Table8 also showed that SUVmax was indipendent prgnostic factor but not Tumor grade in DDLS. Based on the mentioned above, we considered that SUVmax was stronger prognostic factor compared to tumor grade for DDLS in RPS. Those findings were reported by Brenner, Winfried, et al. ("Risk assessment in liposarcoma patients based on FDG PET imaging." European journal of nuclear medicine and molecular imaging 33.11 (2006): 1290-1295.) Those findings was one of the most important points.

We changed the sentence as followings; "These results indicated that SUVmax had the potential clinical usefulness in better assessment of the prognosis for RPS patients with DDLPS and grade 2 tumors." in line 281-282. And we also added the sentence and reference in line 329-331 and reference 20.

3.) the authors mention also the seemingly superiority of the SUVmax as opposed to the widely accepted nomogram. It is kindly suggested to put the own results into more gently perspectives. SUVmax for DDLS may be considered as additional parameter within the nomogram etc. But not more. After all, the authors have NOT shown the results for the stated superiority.

It is as you pointed out.  I am very sorry to have caused your misunderstanding. We did not consider that SUVmax is better than the established nomograms. We consider that the nomograms would be better with SUVmax. We changed the sentence according to your advice.

We change the sentence for "Therefore, SUVmax for grade2 RPS and DDLPS may be considered as additional parameter within the nomogram, resulting in prognostic implications that could help in decision-making regarding treatment intensity, such as combined resection of organs for complete resection of surgery, dose intensity, or regimens for chemotherapy or radiotherapy." in line 331-335.

4.) the expression "should be considered" in the abstract as well as in the conclusion is too strong. It is suggested to use "may be considered".

Thank you for your comment. I corrected it as you pointed out.

We change the word from "should be considered" in the abstract and conclusion to "may be considered".

Reviewer 2 Report

In their manuscript entitled “18F-Fluorodeoxyglucose Positron Emission Tomography is useful in the evaluation of prognosis in retroperitoneal sarcoma”, Wakamatsu and colleagues analyze the applicability of the maximum standardized uptake values (SUV max) of FDG determined by PET as prognosis marker in a series of patients with retroperitoneal soft tissue sarcoma (RPS). The authors demonstrate that a SUVmax higher than 4 is associated with a lower survival of the RPS patients regardless of the histological subtype of the sarcoma. Regarding the histological subtype, SUV max was mainly low in well-differentiated liposarcoma and high in leiomyosarcoma. Additionally, low SUV max was associated with low grade sarcoma, while high grade sarcoma (Grade 3) almost all exhibited higher SUV max. When analyzing the different histological subgroups independently, SUV max was especially a prognostic marker in DDLPS. The authors conclude that FDG-PET might be promising for assessing the prognosis of RPS patients, especially whose with DDLPS.

The manuscript is well written, and the applied methodology as well as the statistics are sound. The conclusions are – at least partly – covered by the results. However, I have some concerns on the manuscript the authors should refer to.

Here are my critics in detail:

  • Some inconsistencies in the Introduction section: for instance, in l. 45 – 52 it should be referred to that this procedure is standard and was not only applied to the analyzed patients cohort. I would recommend switching the wording for instance as follows: “Furthermore, the clinical tumor stage and size is evaluated and then the treatment strategy is designed. However, the aggressiveness of RPS before treatment can not be fully assessed, because biopsy is often not even performed. However, if biopsy was performed, the biopsy tissue sometimes does not reflect the tumor aggressiveness…” Furthermore, the aim of the study (l.71 – 73) is formulated unclear (usefulness of FDG-PET in the clinical assessment?), this should be sharpened.
  • 80: RPS without a pathological diagnosis – what should this mean? Without Grade and/or size? L.82: sexgender? L.99: between 5 cm and 15 cm; Table 1: abbreviation NE should be explained in the footer; l.124: the 3-year DFS rates are given, while in l.137, the 5-year DFS rates are given. This makes it hard to compare the different percentages in histologial subentities and SUV max.
  • My biggest concern: the authors state (for instance in l. 148), that SUV max is a prognostic factor independent of tumor grade. To my opinion, this is not correct. As SUV max is highly associated to tumor grade in grade 1 and 3 RPS, it seems to be only a relevant prognostic factor in grade 2 tumors. Furthermore, lateron (l. 181), the author demonstrate that SUV max is only a prognostic factor in DDLPS (I here leave aside the group of “other” RPS, as this seems to be a highly heterogeneous group of sarcoma, which valid conclusions are hard to draw from – if applicable, it may be interesting to specify these group in a supplemental file). L. 203 – 205 is also questionable in this context. To sum this up, I only see the potential clinical usefulness in an better assessment of the prognosis for RPS patients with DDLPS and grade 2 tumors (which is an important result for this patient group) – this should be better  described in the results, but also in the conclusion section.
  • The discussion section is to me the weakest chapter in the manuscript. L. 217 – 229 seem to be better suited for the introduction. L. 244 – 246 should be omitted from the manuscript. There are no relevant references to already performed studies regarding the prognostic relevance of FDG-PET – there should be at least some discussion on previous studies in sarcoma species (for instance, Subramaniam et al., J Surg Oncol 2021 and Sambri et al., Nucl Med Commun 2019, these two are examples, but not exclusive). L. 274: statistical analyses on LMS have been performed in the manuscript, did you mean RMS?

Finally, I want to thank the authors for sharing their work with the scientific community. Best regards.

Author Response

Dear Reviewer2

Thank you very much for reviewing our manuscript and offering valuable advice.

We have addressed your comments with point-by-point responses and revised the manuscript accordingly.

I attached table of responses for the comments.

1.) l. 45 – 52 it should be referred to that this procedure is standard and was not only applied to the analyzed patient’s cohort. I would recommend switching the wording for instance as follows: “Furthermore, the clinical tumor stage and size is evaluated and then the treatment strategy is designed. However, the aggressiveness of RPS before treatment cannot be fully assessed, because biopsy is often not even performed. However, if biopsy was performed, the biopsy tissue sometimes does not reflect the tumor aggressiveness…

Thank you for your kind comment. We changed the sentences for your recommendation.

We changed the sentence as followings; "Furthermore, the clinical tumor stage and size is evaluated and then the treatment strategy is designed. However, the aggressiveness of RPS before treatment cannot be fully assessed, because biopsy is often not even performed. However, if biopsy was performed, the biopsy tissue sometimes does not reflect the tumor aggressiveness…" in line 66-70.

2.) the aim of the study (l.71 – 73) is formulated unclear (usefulness of FDG-PET in the clinical assessment?), this should be sharpened.

Thank you for your indication. We change the sentence of the aim.

We changed the sentence as followings; "We also investigated the involvement of SUVmax in the clinical outcome of RPS patients compared with other candidate prognostic factors." in line 93-94.

3.) 80: RPS without a pathological diagnosis – what should this mean? Without Grade and/or size?

Thank you for your comment. "Without pathological diagnosis" is meaning that "without biopsy for pathological diagnosis". I am very sorry to have caused your misunderstanding. We change the words.

We changed the words as followings; "without biopsy for pathological diagnosis" in line 101.

4.) L.82: sexgender?

Thank you for your comment. I corrected it as you pointed out.

We changed the words as followings; "gender" in line 119.

5.) L.99: between 5 cm and 15 cm

Thank you for your comment. I corrected it as you pointed out.

We changed the word as followings; "between 5 cm and 15 cm" in line 138.

6.) Table 1: abbreviation NE should be explained in the footer

Thank you for your comment. I added it as you pointed out.

We added the explanation of NE in Table1 as followings; NE; not estimated.

7.) l.124: the 3-year DFS rates are given, while in l.137, the 5-year DFS rates are given. This makes it hard to compare the different percentages in histologial subentities and SUV max.

Thank you for your kind comment. We demonstrated the 5-year DFS rates in line 124.

We changed the data for 5-year DFS rates as followings; " the 5-year DFS rates were 35.9%, 19.9%, 20.5%, and 42.2%..." in line 162.

8.) the authors state (for instance in l. 148), that SUV max is a prognostic factor independent of tumor grade. To my opinion, this is not correct. As SUV max is highly associated to tumor grade in grade 1 and 3 RPS, it seems to be only a relevant prognostic factor in grade 2 tumors.

Thank you for your indication, and I am very sorry to have caused your misunderstanding. In Figure 2c, we demonstrated that grade2 RPS with SUVmax<4 showed better prognosis than that of grade2 or 3 RPS with SUVmax>4. And mutivariate analysis in Table4 showed that SUVmax was indipendent prgnostic factor in RPS. Based on the mentioned above, we considered that SUVmax was stronger prognostic factor for RPS. Those findings was one of the most important points. Therefore, we change the sentence about Figure 2c.

We changed the sentence as followings; "These results suggested that SUVmax was a strong prognostic factor of at least grade2 tumors in RPS." in line 197-198. And we also added the sentence and reference in line 329-331 and reference 20.

9.) lateron (l. 181), the author demonstrate that SUV max is only a prognostic factor in DDLPS (I here leave aside the group of “other” RPS, as this seems to be a highly heterogeneous group of sarcoma, which valid conclusions are hard to draw from – if applicable, it may be interesting to specify these group in a supplemental file).

Thank you for your opinion. We move data of "other" to supplymental data.

We moved Table 6 and Table 9 to supplemental table 1 and 2. We also moved Figure 4b to supplemental Figure 1.

10.) L. 203 – 205 is also questionable in this context. To sum this up, I only see the potential clinical usefulness in an better assessment of the prognosis for RPS patients with DDLPS and grade 2 tumors (which is an important result for this patient group) – this should be better described in the results, but also in the conclusion section.

Thank you for your kind comment. We changed the sentences according to your recommendation.

We changed the sentence as followings; "These results indicated that SUVmax had the potential clinical usefulness in better assessment of the prognosis for RPS patients with DDLPS and grade 2 tumors." in line 280-282.

11.) L. 217 – 229 seem to be better suited for the introduction.

Thank you for your comment. I corrected it as you pointed out.

We modified indicated sentences and moved them to introduction in line 56-63 and 90-91.

12.) L. 244 – 246 should be omitted from the manuscript.

Thank you for your comment. I corrected it as you pointed out.

We omitted the setence in line 337.

13.) There are no relevant references to already performed studies regarding the prognostic relevance of FDG-PET – there should be at least some discussion on previous studies in sarcoma species (for instance, Subramaniam et al., J Surg Oncol 2021 and Sambri et al., Nucl Med Commun 2019, these two are examples, but not exclusive).

Thank you for your indication. We added the refferences and some sentences in discussion.

We added indicated studies in discussion and references in line 312-320 and reference 16 and 17.

14.) L. 274: statistical analyses on LMS have been performed in the manuscript, did you mean RMS?

Thank you for your comment. LMS (Leiomyosarcoma) is collect. In LMS of our study, SUVmax high tended to show worse prognosis compared to SUVmax low, but it was not significant. Therefore, we would like to indicated that we may be able to show the significant difference with more patients' data in LMS.

We did not change the word because of indicated reasons.

Round 2

Reviewer 2 Report

The authors responded to all my points. I do not have further remarks on this manuscript.